## PERSPECTIVES

# Tail of two fishies: age and afferents influence zebrafish lateral-line hair cell regeneration

Lavinia Sheets[1,2] ID

[1]*Department of Otolaryngology, Washington University School of Medicine, St Louis, MO, USA*
[2]*Department of Developmental Biology, Washington University School of Medicine, St Louis, MO, USA*

Email: sheetsl@wustl.edu

Edited by: Ian Forsythe & Corne Kros

Linked articles: This Perspectives article highlights an article by Hardy *et al*. To read this paper, visit https://doi.org/10.1113/JP281522.

The peer review history is available in the Supporting Information section of this article (https://doi.org/10.1113/JP282012#support-information-section).

Hair cells are exquisitely sensitive mechano-receptors that mediate the senses of hearing, balance and spatial orientation in vertebrates. Environmental stressors such as noise, ageing and certain antibiotics and chemotherapeutic drugs contribute to irreversible loss of hair cells, resulting in permanent hearing loss and/or balance disorders. In aquatic vertebrates, hair cells are also found in the lateral-line system, which is a sensory system used to detect fluid motion in the animal's surrounding aqueous environment. In contrast to mammals, aquatic vertebrates are capable of fully regenerating hair cells. Previous studies in larval zebrafish have characterized the morphological recovery of lateral-line hair cells following ototoxic drug-induced hair cell death and have shown hair cell regeneration is rapid, with complete recovery of hair cell number within 48–72 h. Additional investigations have revealed that specific populations of non-sensory supporting cells proliferate and give rise to newly formed hair cells (Lush *et al*. 2019; Thomas & Raible, 2019).

In addition to evoked activity in response to sensory stimuli, afferent neurons of the lateral line organ generate spontaneous action potentials, and this spontaneous activity is driven by calcium-dependent neurotransmitter release from

hair cells (Trapani & Nicolson, 2011). While morphological regeneration of zebrafish lateral-line hair cells has been well studied, the functional recovery of lateral-line organs – that is to say, the ability of regenerated hair cells to induce spontaneous and evoked activity in innervating afferents nerves over the time course of regeneration – had not previously been defined. Further, as most research on zebrafish lateral-line regeneration is performed on younger larvae with functional but still immature hair cells, it was also unclear how developmental age influenced both morphological and functional hair cell regeneration.

In this issue of *The Journal of Physiology*, Hardy *et al*. (2021) address these questions by examining lateral-line regeneration during two stages of larval development: early stage (<8 days post-fertilization) in which lateral-line hair cells drive spontaneous and evoked firing of innervating afferent nerves but display immature biophysical properties, and late-stage (12–17 days post-fertilization), in which lateral-line organs contain mature hair cells (Olt *et al*. 2014). To investigate regeneration of profoundly damaged lateral-line organs, the authors used copper to induce near-total hair cell death. In early-stage larvae, and consistent with previous studies, morphological recovery of hair cell number was observed within 48–72 h. Hardy *et al*. then went further to define functional recovery of regenerated hair cells by assessing spontaneous and evoked lateral-line activity. In early-stage larvae, signs of functional recovery were apparent well before full morphological recovery; spontaneous afferent action potentials returned within ∼20 h following treatment. In addition, spontaneous hair cell neurotransmitter release, measured using a genetically encoded reporter of glutamate (iGluSnFR), was detected in approximately half of regenerated hair cells within 36–48 h, indicating hair cells were at various stages of maturation during regeneration. Employing fluid-jet stimulation of hair cells to examine evoked activity, Hardy *et al*. verified that evoked spiking activity of afferent neurons completely recovers in early-stage larval zebrafish by 48 h, and this functional recovery appeared to correspond with a period of morphological synaptic

refinement between 24 and 48 h following copper treatment. Correspondingly, fluid jet-evoked calcium responses in regenerating hair cells and afferent terminals showed progressive recovery. These data cumulatively reveal a gradual process of morphological and functional recovery in early-stage larvae following copper-induced lateral-line ablation whereby regenerated hair cells can drive action potentials as early as 24 h post-treatment but require 48 h for full recovery. Notably, in late-stage larvae with more mature lateral-line organs, Hardy *et al*. observed the time scale of complete morphological and functional regeneration was much slower (5 days), but evoked hair cell and afferent terminal calcium responses were comparable to untreated fish by 48 h post-treatment. From the broader perspective on the impact of sensory system damage to fish behaviour, the restoration of partial function before full recovery of lateral-line organs may be important to restore some lateral line-mediated behaviours needed for survival, such as predator avoidance.

Hardy *et al*. also report the intriguing observation that afferent terminals appear to contact supporting cells during the early stages of regeneration and speculate that afferents may also play a key role in hair cell regeneration. This hypothesis was tested by using two-photon illumination to completely ablate the lateral-line ganglion, then quantifying regeneration after treatment with a high concentration of copper. Although the authors observed deficits in hair cell regeneration after ganglion ablation, it is noteworthy that they also observed impaired recovery of supporting cells, which serve as progenitors to regenerated hair cells. Nevertheless, these results suggest that afferents may influence the process of hair cell regeneration and should stimulate further research to determine whether afferents also regulate regeneration in the vertebrate inner ear. Finally, the authors note that ablation of the lateral line ganglion causes a progressive loss of hair cells and supporting cells, even without ototoxic damage. Similar findings have been reported for the mammalian inner ear (Favre & Sans, 1991) and suggest that afferent innervation plays an important role in the maintenance of sensory hair cells.

The Journal of Physiology

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

## Additional information

### Competing interests

None.

### Author contributions

Sole author.

### Funding

None.

### Acknowledgements

The author thanks Mark Warchol for feedback and editorial comments.

### Keywords

afferent fibres, electrophysiology, hair cells, regeneration, zebrafish

### Supporting information

Additional supporting information can be found online in the Supporting Information section at the end of the HTML view of the article. Supporting information files available:

**Peer Review History**

