## [Peer Review History · The Journal of Physiology]

□

Tail of two fishies: age and afferents influence zebrafish lateral-line hair cell regeneration

Lavinia Sheets

DOI: 10.1113/JP282012

Corresponding author(s): Lavinia Sheets (sheetsl@wustl.edu)

The following individual(s) involved in review of this submission have agreed to reveal their identity: Walter Marcotti (Referee #1)

Review Timeline:

Submission Date:

01-Jul-2021

Accepted:

09-Jul-2021

Senior Editor: Ian Forsythe

Reviewing Editor: Corne Kros

Transaction Report:

Dear Dr Sheets,

Re: JP-P-2021-282012 "Tail of two fishies: age and afferents influence zebrafish lateral-line hair cell regeneration" by Lavinia Sheets

I am pleased to tell you that your invited Perspectives article has been accepted for publication in The Journal of Physiology.

NEW POLICY: In order to improve the transparency of its peer review process The Journal of Physiology publishes online as supporting information the peer review history of all articles accepted for publication. Readers will have access to decision letters, including all Editors' comments and referee reports, for each version of the manuscript and any author responses to peer review comments. Referees can decide whether or not they wish to be named on the peer review history document.

The last Word version of the paper submitted will be used by the Production Editors to prepare your proof. When this is ready you will receive an email containing a link to Wiley's Online Proofing System. The proof should be checked and corrected as quickly as possible.

All queries at proof stage should be sent to tjp@wiley.com

Thank you very much for your contribution to The Journal of Physiology.

Yours sincerely,

Ian D. Forsythe
Deputy Editor-in-Chief
The Journal of Physiology
<https://jpp.msubmit.net>
<http://jpp.physoc.org>
The Physiological Society
Hodgkin Huxley House
30 Farringdon Lane
London, EC1R 3AW
UK
<http://www.physoc.org>
<http://journals.physoc.org>

EDITOR COMMENTS

Reviewing Editor:

This is an excellent perspective, written in an accessible style and highlighting key findings from the paper.

Senior Editor:

Many thanks for an interesting perspective.

Confidential Review

01-Jul-2021